# Grasp Proposal Networks: An End-to-End Solution for Visual Learning of Robotic Grasps

Chaozheng Wu[1*]   Jian Chen[1*]   Qiaoyu Cao[1]   Jianchi Zhang[1]

Yunxin Tai[1]   Lin Sun [2]   Kui Jia[1†]

[1]South China University of Technology       [2]Samsung, USA
{eeczwu,ee_chenjian,eeqycao,msj.c.zhang}@mail.scut.edu.cn
yunxintai@gmail.com   lin1.sun@samsung.com   kuijia@scut.edu.cn

## Abstract

Learning robotic grasps from visual observations is a promising yet challenging task. Recent research shows its great potential by preparing and learning from large-scale synthetic datasets. For the popular, 6 degree-of-freedom (6-DOF) grasp setting of parallel-jaw gripper, most of existing methods take the strategy of heuristically sampling grasp candidates and then evaluating them using learned scoring functions. This strategy is limited in terms of the conflict between sampling efficiency and coverage of optimal grasps. To this end, we propose in this work a novel, end-to-end *Grasp Proposal Network (GPNet)*, to predict a diverse set of 6-DOF grasps for an unseen object observed from a single and unknown camera view. GPNet builds on a key design of grasp proposal module that defines *anchors of grasp centers* at discrete but regular 3D grid corners, which is flexible to support either more precise or more diverse grasp predictions. To test GPNet, we contribute a synthetic dataset of 6-DOF object grasps; evaluation is conducted using rule-based criteria, simulation test, and real test. Comparative results show the advantage of our methods over existing ones. Notably, GPNet gains better simulation results via the specified coverage, which helps achieve a ready translation in real test. Our code and dataset are available on `https://github.com/CZ-Wu/GPNet`.

## 1   Introduction

Robotic object grasping is one of the basic functions that a robot system aims to emulate our human beings. The task is challenging due to imprecision in sensing, planning, and actuation, and also due to the possible absence of knowledge about physical properties of the object (e.g., mass distribution and surface material). It was recently demonstrated that deep learning on annotated datasets of robotic grasp can achieve good robustness and generalization [13, 21, 14, 10, 11]. Methods based on synthetic data (e.g., object CAD models and the correspondingly rendered images) [17, 5, 32] show particular promise, as they can ideally generate as many as infinite numbers of grasp annotations. Even though there exists a risk of domain discrepancy between simulated and real environments, deep learning models trained on such synthetic datasets show remarkable performance on real-world grasp testings, with better generalization to novel object instances and categories [17, 5]. In this work, we study deep learning optimal grasp configurations from synthetic images, with a particular focus on grasping with a parallel-jaw gripper, whose parametrization is typically of 6 degrees of freedom (6-DOFs), including 3D gripper center of the grasping location and 3D gripper orientation.

---

[*]Authors contributed equally.
[†]Corresponding author.

Optimal grasp configurations depend on working conditions in real-world environments. In many cases, due to kinematical constraints of robotic arms and/or possible collisions, a diverse set of multiple grasp predictions are expected such that there exist grasps among the predictions that can be successfully actuated. There generally exist two strategies to predict multiple grasps for a given object. The first strategy is used in [29, 30], which samples grasp candidates from observed object surface via heuristic manners, and then evaluates them using learned scoring functions; alternatively, full object surface model is assumed in [32, 15] to sample more reliable grasp candidates during the test phase. The second strategy learns to directly predict multiple grasps. We argue that the first strategy is limited in the following aspects: (1) sampling can only be made finite, making it possible to miss optimal grasps, (2) increasing the density of grasp candidate sampling increases linearly the computation costs of both the sampling itself and the subsequent grasp estimation — note that sampling itself costs significantly [29]. To address the limitation, a first attempt is made in [20] that learns a latent grasp space via variational auto-encoder (VAE), and promising grasps can be obtained by sampling from the learned latent space. However, we empirically find that grasps given by the VAE model of [20] tend to focus on a single mode, e.g., centers of their predicted grasps are close to the object mass center; in order words, their generated grasps are less diverse.

In this work, we propose a novel end-to-end solution of *Grasp Proposal Network (GPNet)*, in order to predict a diverse set of 6-DOF grasps for an unseen object observed from a single and unknown camera view. Figure 1 illustrates our pipeline. GPNet builds on a key design of grasp proposal module that defines *anchors of grasp centers* at a discrete set of regular 3D grid corners. It stacks three headers on top of the grasp proposal module, which for any grasp proposal, are trained to respectively specify antipodal validity [2], regress a grasp prediction and score the confidence of final grasp. The proposed GPNet is in fact flexible enough to support either more precise or more diverse grasp predictions, by focusing or spreading the anchors of grasp centers in the 3D space. To test GPNet, we contribute a synthetic dataset of 6-DOF object grasps, including $22.6M$ annotated grasps for 226 object models. We evaluate our proposed GPNet in terms of rule-based criteria, simulation test, and also real test. Experiments show the advantages of our method over existing ones.

## 2   Related Works

**Grasp Annotations and Datasets** Existing datasets for robotic grasp learning are annotated based on four types: (1) human labeling by either grasping objects in real environments [13, 12] or by demonstrating grasps in simulation engines [32], (2) automating physical grasp trials by robots [21, 14, 8], (3) analytic computation of grasp quality metrics [18, 17], and (4) automating simulation of physical grasps in physics engines [5, 32]. The last two approaches are related to this work by annotating on synthetic data. Particularly, Dex-Nets [18, 17] compute analytic grasp qualities to obtain millions of annotations over more than $10K$ object models, and Jacquard [5] simulates grasp trials in physics engine to obtain a similar amount of annotations. However, their annotations are only of simplified planar grasps. Recent works [20, 15] present synthetic datasets of 6-DOF grasps, whose testing environments are not publicly available yet to benchmark different methods.

**Deep Visual Grasp Learning** Given availability of synthetic grasp datasets, deep grasp learning is drawing attention recently in both robotics and vision communities. Borrowing ideas from 2D object detection [25, 24, 16], Jiang *et al.* propose 2D oriented rectangles to represent simplified grasps, and Chu *et al.* [3] re-cast angle regression as classification by discretizing the space of in-plane rotation and then utilizing Faster-RCNN [25] to detect multiple grasp poses from a single RGB-D image. A generative approach is also proposed in [19] to directly predict a grasp pose at each pixel of feature maps via fully-convolutional network. Redmon and Angelova [23] propose a one-stage method to directly regress grasp poses, similar to [24]. Florence *et al.* [7] propose to transfer grasps across different objects based on the approaches of dense correspondence [27, 9, 33]. To take the full object surface into account, Yan *et al.* [32] and Merwe *et al.* [31] present geometry-aware grasp evaluation methods respectively via 3D occupancy grid and signed distance function.

## 3   The Problem of Visual Grasp Learning

We consider in this work an ideal but common setting of grasping a singulated object resting on a plane of table using an end-effector of parallel-jaw gripper. The problem concerns with estimation of parameterized grasp poses in a 3D coordinate space.

## 3.1 Parameterizations and Learning

Given an object $\mathcal{O}$ with its mass center $\mathbf{z} \in \mathbb{R}^3$, denote 3D shape of the object surface as $\mathcal{S}$. Let $\mathbf{z}$ be the origin of world coordinate system. A grasp based on parallel-jaw gripper can be parameterized as $\mathbf{g} = (\mathbf{x}, \theta) \in SE(3)$, where $\mathbf{x} = (x, y, z) \in \mathbb{R}^3$ locates the center of two parallel jaws, $\theta \in [-\pi, \pi]^3$ is the Euler angle vector representing 3D orientation of the gripper — we note that an additional freedom of gripper opening width $w \in \mathbb{R}^+$ is sometimes used in the literature. An illustration of such a 6-DOF grasp parameterization is given in the supplementary material. Euler angle representation of 3D pose is physically intuitive but disadvantageous in that it has singularities when implementing rotations; in this work, we implement 3D orientation of a grasp pose using unit quaternion [1]. Other than grasp parameterization, success or failure of a grasp also depends on physical properties of the object, such as mass distribution and surface material. For simplicity, we assume in this work a fixed friction coefficient $\gamma$ for the surface material, and that the mass center $\mathbf{z}$ coincides with geometric center of the shape $\mathcal{S}$. We do not consider the uncertainty when measuring $\mathcal{S}$ and $\mathbf{g}$.

Consider a vision-guided robotic grasp scenario where a camera points towards the grasp environment (e.g., centroid of the object). The objective of visual grasp learning is to estimate from visual observations optimal grasp configurations that specify where and how to grasp the object. For a depth camera, denote the point cloud representation of the visible surface of an object $\mathcal{O}$ as $\mathcal{I} = \{\mathbf{p}_i \in \mathbb{R}^3\}_{i=1}^n$, where $n$ is the number of observed points and $\mathbf{p}_i$ contains the 3D coordinates of the $i^{th}$ point. Depending on availability of grasp candidates $\{\mathbf{g} \in \mathcal{G}\}$, which can be sampled from the object surface $\mathcal{S}$ as described shortly, the task of visual grasp learning can be formalized as either learning a scoring function

$$\Phi : \mathbb{R}^{n \times 3} \times (\mathbb{R}^3 \times [-\pi, \pi]^3) \to [0, 1], \tag{1}$$

which ranks $\{\mathbf{g} \in \mathcal{G}\}$ to specify an optimal grasp [17], or learning a regression function

$$\Psi : \mathbb{R}^{n \times 3} \to \mathbb{R}^3 \times [-\pi, \pi]^3, \tag{2}$$

which directly estimates one or multiple grasps [5]. To evaluate any estimated $\hat{\mathbf{g}} = (\hat{\mathbf{x}}, \hat{\theta})$, we consider in this work the following three criteria.

**Rule-based evaluation** Given ground-truth positive grasps $\{\mathbf{g}^{*+} = (\mathbf{x}^{*+}, \theta^{*+}) \in \mathcal{G}^{*+}\}$ annotated on $\mathcal{O}$, $\hat{\mathbf{g}}$ is counted as a *success* if it satisfies the conditions of $\|\hat{\mathbf{x}} - \mathbf{x}^{*+}\|_2 \leq \Delta_{\mathbf{x}}$ and $\|\hat{\theta} - \theta^{*+}\|_\infty \leq \Delta_\theta$ for any one of $\{\mathbf{g}^{*+} \in \mathcal{G}^{*+}\}$.

**Evaluation via simulation** For any estimated $\hat{\mathbf{g}}$, we conduct a simulation whose environment is specified shortly in Section 3.2. It is counted as a *success* if the simulated gripper can lift the object using $\hat{\mathbf{g}}$ to a certain height and stably move it around.

**Real test** For any estimated $\hat{\mathbf{g}}$, robot agents the top confident grasp without physics violation. Once it elevates objects over 30cm and returns to original state, this run is treated as a *success*.

As indicated by functions (1) and (2), the nature of visual grasp learning is to build mapping relations from observed shape geometries of object surface to physically (and semantically) sensible grasp configurations. The task is challenging due to the following factors: (1) physical properties such as mass distribution and surface material are usually unavailable; (2) for a given object, optimal grasps are not uniquely defined, and the ambiguity is even worse when considering the gap between physical/geometric and semantic grasp annotations; (3) grasps of an object can only be annotated up to a discrete and possibly sparse set, which causes difficulties for both training and evaluation of visual grasp learning; (4) there exists an issue of transferability from grasp estimations learned from synthetic data to use in real-world testing. Nevertheless, promising results in recent works [17, 5, 32, 20] demonstrate the advantage of visual grasp learning over the traditional pipeline composed of separate steps of object detection, segmentation, registration, and pose estimation. We thus aim for taking visual grasp learning as an isolated machine learning task.

## 3.2 Synthetic Dataset Construction

We summarize our synthetic dataset of object grasps that is constructed using physics engine of PyBullet [4]. More details of the dataset acquisition framework can be found in the supplemental material. Our dataset is based on ShapeNetSem [26], which contains annotations of physical attributes (e.g., material density and static friction coefficient) essential for our grasp annotation. We use 226

CAD models of 8 categories (*bowl, bottle, mug, cylinder, cuboid, tissue box, sodacan,* and *toy car*) in ShapeNetSem as our models of interest for simulated grasps. We totally obtain $22.6M$ grasp annotations ($\sim 100,000$ per object), of which $\sim 23.6\%$ are positive annotations and $\sim 76.4\%$ are negative ones. For each grasp candidate, we also compute analytic quality score based on [6]. To prepare inputs of visual grasp learning, we render RGB-D images under 1000 arbitrary views for each object model. To use the dataset, we split object models (of all categories) into *training* set (196 objects) and *test* set (30 objects). This is to prepare a testing scenario of generalization to novel object instances.

# 4   Grasp Proposal Networks

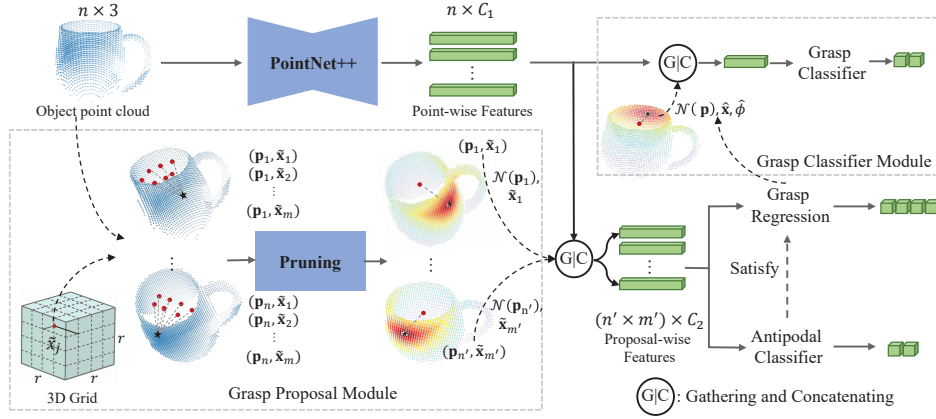

Figure 1: Overview of our GPNet architecture. Given a point cloud $\mathcal{I}$ of partial object surface with $n$ points. GPNet uses a backbone network of PointNet++ to extract point-wise features. In the Grasp Proposal module, $m = r^3$ anchors $\{\tilde{x}_j\}_{j=1}^m$ of grasp centers are defined at a discrete set of regular 3D grid. For each $p_i$ in $\mathcal{I}$, we connect it with all anchors and get a set of grasp proposals $\mathcal{G}_i = \{(p_i, \tilde{x}_j)\}_{j=1}^m$, resulting in a number of $n \times m$ grasp proposals in total. In this work, two physically sensible schemes are designed to prune most of the $n \times m$ grasp proposals to a number of $n' \times m'$. With these grasp proposals, we gather the features from PointNet++ associated with the anchor coordinates to predict a diverse set of precise grasps using three headers, which are trained to respectively specify antipodal validity, regress grasp prediction, and score grasp confidence.

To implement visual grasp learning, we propose in this work a Grasp Proposal Network (GPNet) that aims to predict a diverse set of 6-DOF grasps $\{\hat{g}\}$ for an object $\mathcal{O}$, given a point cloud $\mathcal{I}$ of partial object surface observed from a single and unknown camera view. Our GPNet thus learns an instantiation of the function (2). In GPNet, we use a re-parametrization $(c_1, c_2, \phi) \in \mathbb{R}^7$ of $g = (x, \theta)$ (an illustration can be found in the supplemental material), where $(c_1, c_2)$ denote the two contact points on the surface, which already determine the grasp center $x = (c_1 + c_2)/2$, and the "roll" and "yaw" orientations of $\theta$, and $\phi$ denotes the left freedom of "pitch" orientation [3]. Replacing $c_2$ with the center $x$ gives an equivalent parametrization $g = (c_1, x, \phi)$.

Given the new grasp parameterizations, GPNet builds on a key idea of defining *anchors of grasp centers* at a discrete set of regular 3D grid positions $\{\tilde{x}_j\}_{j=1}^m$, which together with each $p_i$ of observed surface points $\{p_i\}_{i=1}^n$, give a set $\mathcal{G}_i$ of grasp proposals $\{(p_i, \tilde{x}_j)\}_{j=1}^m$ short of the freedom of angle $\phi$; we train the GPNet to predict an angle $\hat{\phi}$ and an offset $\Delta_{\tilde{x}_j} \in \mathbb{R}^3$ for each $\tilde{x}_j$. Our scheme supports natural variants that favor either more precise or more diverse grasp proposals, by focusing or spreading the grid positions $\{\tilde{x}_j\}_{j=1}^m$ in the 3D space, as presented shortly in Section 4.1. For any observed $p_i$, we learn a feature $f_i$ based on points of $\{p_i\}_{i=1}^n$ in its local neighborhood via an anchor-dependent fashion, which together with coordinates of an anchor $\tilde{x}_j$, form features of the grasp proposal $(p_i, \tilde{x}_j)$ .

As illustrated in Figure 1, our proposed GPNet stacks three headers on top of the grasp proposal module and PointNet++ based feature extractor [22]. For any input proposal $(\mathbf{p}_i, \widetilde{\mathbf{x}}_j)$, the three headers are trained to respectively specify its antipodal validity [2], regress a grasp prediction $\hat{\mathbf{g}} = (\mathbf{p}_i, \hat{\mathbf{x}} = \widetilde{\mathbf{x}}_j + \Delta_{\widetilde{\mathbf{x}}_j}, \hat{\phi})$, and score confidence of $\hat{\mathbf{g}}$. Details of our contributed components in GPNet are specified as follows.

## 4.1 Diverse and Flexible Grasp Proposals via 3D Grid Anchors

Due to kinematical constraints of robotic arms and/or possible collisions in working environments, grasp predictions are in many cases expected to be diversified, such that there exist grasps among the predictions that can be successfully actuated. Our grasp proposal module in GPNet is specially designed for this purpose. Inspired by anchor based 2D object detections [25, 16], we take into account the physical nature of 6-DOF grasps and define *anchors of grasp centers* at regular grid corners in a normalized 3D space, whose size is assumed to fit in the objects of interest. Figure 1 gives an illustration. Specifically, we evenly partition the normalized 3D space into $r \times r \times r$ cells at $m = r^3$ grid corners, giving rise to the anchors $\{\widetilde{\mathbf{x}}_j\}_{j=1}^m$. A set $\mathcal{G}_i$ of grasp proposals $\{(\mathbf{p}_i, \widetilde{\mathbf{x}}_j)\}_{j=1}^m$ are formed by connecting each $\mathbf{p}_i$ in observed surface points $\{\mathbf{p}_i\}_{i=1}^n$ with the $m$ anchors. Such proposals are short of the freedom of angle $\phi$; we train the GPNet to predict an angle $\hat{\phi}$ and an offset $\Delta_{\widetilde{\mathbf{x}}_j} \in \mathbb{R}^3$ for each $\widetilde{\mathbf{x}}_j$, as described shortly in Section 4.4.

The total number of grasp proposals defined above could be as large as $n \times m$, we design in GPNet two physically sensible schemes to efficiently prune them during training and/or test phases. Since the object surface model $\mathcal{S}$ is available during training, our first scheme simply removes those grid corners falling outside of $\mathcal{S}$. Our second scheme relies on the antipodal constraint of contact points [2]. In the training phase, given the annotated ground-truth grasps $\{\mathbf{g}^* \in \mathcal{G}^*\}$, we only keep those proposals in $\{\mathcal{G}_i = \{(\mathbf{p}_i, \widetilde{\mathbf{x}}_j)\}_{j=1}^m\}_{i=1}^n$ that are close to any $\mathbf{g}^*$, and label them as positive samples of antipodal validity, while dropping other proposals from which we also sample a fixed subset as negative samples of antipodal validity; we train a classifier (the first header of GPNet) to tell validity of antipodal constraint for any grasp proposal in the test phase, in order to improve the inference efficiency. Note that all ground-truth grasp annotations satisfy antipodal constraint. We empirically find that our schemes remove at least $86.4\%$ of the total $n \times m$ grasp proposals during training.

**Variants for Precise or Diverse Proposals** We note that for each anchor $\widetilde{\mathbf{x}}_j$, the range of offset value $\Delta_{\widetilde{\mathbf{x}}_j} \in \mathbb{R}^3$ to be regressed depends on the resolution $r$ of grid cells in the 3D proposal space, which in turn determines the precision of regression made by GPNet. Increasing the resolution $r$ would give more precise predictions, however, it increases the number of grasp proposals cubically as $m = r^3$. When grasp diversity is not an issue in some working environments, our proposed module can be adapted by focusing the fixed $m$ number of grid corners close to mass center of the object, which is simply assumed to the origin of the 3D coordinate space in this work, thus potentially improving prediction precision at the sacrifice of reduced diversity.

## 4.2 Extraction of Grasp Features from Anchor-dependent Local Surface Points

For any grasp proposal $(\mathbf{p}_i, \widetilde{\mathbf{x}}_j)$, we use an anchor-dependent manner to determine a local point neighborhood $\mathcal{N}(\mathbf{p}_i)$ around $\mathbf{p}_i$, as illustrated in Figure 2. The neighborhood $\mathcal{N}(\mathbf{p}_i)$ includes some surface points in the observed $\{\mathbf{p}_i\}_{i=1}^n$, based on which we use a backbone network of PointNet++ [22] to extract a feature vector $\mathbf{f}_i$ for $\mathcal{N}(\mathbf{p}_i)$.

Our adaptive neighborhood is aligned with the use of parallel-jaw gripper. Intuitively, the gripper would hold an object most firmly when one jaw of the gripper touches the object surface at $\mathbf{p}_i$ from direction perpendicular to the surface tangent plane at $\mathbf{p}_i$. We thus determine whether to include any $\mathbf{p}_{i'}$ in the neighborhood $\mathcal{N}(\mathbf{p}_i)$ by comparing the angle between vectors $\overrightarrow{\widetilde{\mathbf{x}}_j \mathbf{p}_i} = \mathbf{p}_i - \widetilde{\mathbf{x}}_j$ and $\overrightarrow{\mathbf{p}_i \mathbf{p}_{i'}} = \mathbf{p}_{i'} - \mathbf{p}_i$, together with the distance $d(\mathbf{p}_{i'}, \mathbf{p}_i) = \|\mathbf{p}_{i'} - \mathbf{p}_i\|$, giving the criterion

$$\mathcal{N}(\mathbf{p}_i, \widetilde{\mathbf{x}}_j) = \{\mathbf{p}_{i'} \mid d(\mathbf{p}_{i'}, \mathbf{p}_i) \cdot (|\cos(\overrightarrow{\mathbf{p}_i \mathbf{p}_{i'}}, \overrightarrow{\widetilde{\mathbf{x}}_j \mathbf{p}_i})| + 1) \le \varepsilon\} \tag{3}$$

where $\varepsilon$ is a threshold to be specified. The criterion (3) generally gives an anisotropic neighborhood as shown in Figure 2. We concatenate feature $\mathbf{f}_i$ of $\mathcal{N}(\mathbf{p}_i)$ with the anchor coordinates to form $[\mathbf{f}_i; \widetilde{\mathbf{x}}_j]$ as the feature of grasp proposal $(\mathbf{p}_i, \widetilde{\mathbf{x}}_j)$. Experiments in Section 5 show that anchor coordinates provide additional information for grasp regression.

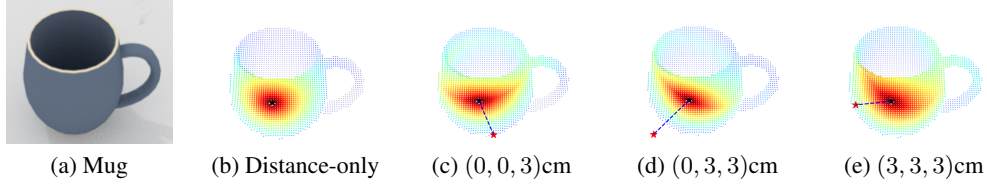

| (a) Mug | (b) Distance-only | (c) $(0,0,3)$cm | (d) $(0,3,3)$cm | (e) $(3,3,3)$cm |

Figure 2: The visualization of black pentagram's neighbor scopes by different means on mug point clouds. (b) is generated by distance-only method. (c)-(e) are generated by the anchor-dependent method, captions below are the coordinates of grids (red pentagrams in figure).

### 4.3 Scoring of Regressed Grasps

A module of grasp classification is essential to score predicted grasps and suggest which ones are most likely to be successfully actuated. In [3, 23, 19], a grasp classifier parallel to grasp regression is designed, which has the shortcoming that the regressed offsets are not considered in grasp scoring. As a result, the output scores are less consistent with the quality of predicted grasps. As a remedy, 6-DOF GraspNet [20] uses an independent grasp evaluator to score the predicted grasps. In this work, we design a header of grasp classification in GPNet that can be trained with other two headers jointly. For each predicted grasp $\hat{\mathbf{g}} = (\mathbf{p}, \hat{\mathbf{x}}, \hat{\phi})$, we first calculate the neighborhood $\mathcal{N}(\mathbf{p})$, and then concatenate the feature $\mathbf{f}$ of $\mathcal{N}(\mathbf{p})$ with the predicted center and angle to form $[\mathbf{f}; \hat{\mathbf{x}}; \hat{\phi}]$, which is used as input of the classifier.

### 4.4 Training and Inference

We present in this section our used loss terms respectively for the three headers of GPNet.

**Antipodal Validity Loss** Grasp proposals may violate antipodal constraint of contact points [2]. To tell the antipodal validity of any grasp proposed in the test phase, we use a binary classifier as the first header of GPNet, and train it using positive and negative examples of antipodal validity specified in Section 4.1. Denote $l^*_{AP}(\mathbf{p}_i, \widetilde{\mathbf{x}}_j)$ as the label of antipodal validity for a training proposal $(\mathbf{p}_i, \widetilde{\mathbf{x}}_j)$, and $\hat{l}_{AP}(\mathbf{p}_i, \widetilde{\mathbf{x}}_j)$ as its prediction, we write the cross-entropy as

$$\mathcal{L}_{AP}(\mathbf{p}_i, \widetilde{\mathbf{x}}_j) = -l^*_{AP}(\mathbf{p}_i, \widetilde{\mathbf{x}}_j) \log \hat{l}_{AP}(\mathbf{p}_i, \widetilde{\mathbf{x}}_j) - (1 - l^*_{AP}(\mathbf{p}_i, \widetilde{\mathbf{x}}_j)) \log \left(1 - \hat{l}_{AP}(\mathbf{p}_i, \widetilde{\mathbf{x}}_j)\right). \quad (4)$$

**Grasp Regression Loss** When a proposal $(\mathbf{p}_i, \widetilde{\mathbf{x}}_j)$ is classified as positive by the first header, we train the second header of GPNet to regress $\Delta_{\widetilde{\mathbf{x}}_j}$ and $\hat{\phi}$, which give the grasp prediction $\hat{\mathbf{g}} = (\mathbf{p}_i, \hat{\mathbf{x}} = \widetilde{\mathbf{x}}_j + \Delta_{\widetilde{\mathbf{x}}_j}, \hat{\phi})$. Based on our pruning scheme in Section 4.1, there could exist multiple ground-truth positive grasps $\{\mathbf{g}^{*+}_k = (\mathbf{p}_i, \mathbf{x}^{*+} = \widetilde{\mathbf{x}}_j + \Delta^{*+}_{\widetilde{\mathbf{x}}_j}, \phi^{*+}_k)\}$ for the proposal $(\mathbf{p}_i, \widetilde{\mathbf{x}}_j)$, since the proposal does not concern with the freedom of "pitch" orientation. To deal with this one-to-many mapping ambiguity, we are inspired by exploitation on policy reward design [28], and use weighted losses for these ground-truth grasps. Assume there exist $K$ such positive grasps, we use the following form of regression loss

$$\mathcal{L}_{REG}(\mathbf{p}_i, \widetilde{\mathbf{x}}_j) = \|\Delta^{*+}_{\widetilde{\mathbf{x}}_j} - \Delta_{\widetilde{\mathbf{x}}_j}\| + \frac{1}{K} \sum_{k=1}^{K} \omega_k |\cos \hat{\phi} - \cos \phi^{*+}_k| \quad (5)$$

where $\omega_k$ is a scalar weight inversely proportional to the magnitude of $|\cos \hat{\phi} - \cos \phi^{*+}_k|$.

**Grasp Classification Loss** Our ground-truth grasps $\{\mathbf{g}^* \in \mathcal{G}^*\}$ on an object $\mathcal{O}$ are annotated by performing simulation in the physics engine. All grasps in $\mathcal{G}^*$ satisfy the antipodal constraint, but some of them are labeled as negative, since they fail to actuate grasp trails due to collision or sliding of the object from the gripper. Let $\mathcal{G}^{*+}$ and $\mathcal{G}^{*-}$ respectively contain the positive and negative grasp annotations. For any regressed grasp prediction $\hat{\mathbf{g}}$, we label it as positive when it is closer to a $\mathbf{g}^{*+} \in \mathcal{G}^{*+}$; otherwise we label it as negative. We use a binary classifier as the third header of GPNet, and train it using the thus obtained positive and negative samples. Denote $l^*_{CLS}(\hat{\mathbf{g}})$ as the label and $\hat{l}_{CLS}(\hat{\mathbf{g}})$ as the prediction, we write the cross-entropy as

$$\mathcal{L}_{CLS}(\hat{\mathbf{g}}) = -l^*_{CLS}(\hat{\mathbf{g}}) \log \hat{l}_{CLS}(\hat{\mathbf{g}}) - (1 - l^*_{CLS}(\hat{\mathbf{g}})) \log \left(1 - \hat{l}_{CLS}(\hat{\mathbf{g}})\right). \quad (6)$$

Combing (4), (6), and (5), we write our overall loss function as

$$\mathcal{L}_{GPNet} = \mathcal{L}_{AP} + \alpha\mathcal{L}_{CLS} + \beta\mathcal{L}_{REG}, \tag{7}$$

where $\alpha$ and $\beta$ are weighting parameters.

## 5 Experiment

**Models and Implementation Details** We use the standard PointNet++ as in [22], which gives point-wise features. Our anchor-dependent neighborhood scheme includes an adaptive size of points, we either upsample or downsample them to a fixed size of 100 points, where we set $\varepsilon$ in (3) as $0.022\sqrt{3}$. The hyper-parameters in objective (7) are set to $\alpha = \beta = 1$ respectively. For each iteration, we feed into the network positive and negative samples according to the ratio of $3 : 7$. In terms of training parameters, all model in our experiments are trained with SGD optimizer using an initial learning rate 0.001, weight decay 0.0001 and momentum 0.9. During test phase, we first select those grasp proposals with positive scores of antipodal classifier, then regress corresponding grasp predictions, and finally rank them according to their scores of confidence from the grasp classifier. We use Non-Maximum Suppression (NMS) to obtain a diverse set of grasp predictions which ideally distribute across the object surface. The NMS settings are as follows: for a reference prediction of higher confidence, we remove its neighboring ones when their center predictions are within 40mm of the reference one and each of their respective 3 angle predictions is within $60°$ of the reference one.

**Evaluation Metrics and Comparisons** We use two metrics of success rate@$k\%$ and coverage rate@$k\%$ to evaluate different models quantitatively. Success rate@$k\%$ and coverage rate@$k\%$ are similar to the metrics of precision@$k$ and recall@$k$ popularly used in retrieval, with the only difference that we consider a ratio of $k\%$ over all the NMS filtered predictions for an object. We set $k = 10, 30, 50, 100$. We compare our proposed GPNet with a naive version, the state-of-the-art 6-DOF GraspNet [20], and also the planar grasp method of GQCNN in DexNet [17]. For GPNet, we also study its variants by setting different values of the resolution $r$ in varying sizes of 3D proposal space. The baseline of GPNet-Naive simply removes the anchor coordinates of grasp centers from features of grasp proposals. In 6-DOF GraspNet, we follow [20] and use latent space dimension of 2. In GQCNN, all hyper-parameters are optimally tuned to have the best performance. For all models, we use training set of our contributed dataset for training, and rule-based and simulation results are reported on the test set. For rule-based evaluation, we set the thresholds as $\Delta_{\mathbf{x}} = 25$mm and $\Delta_{\theta} = 30°$.

### 5.1 Ablation Studies

In Table 1, we report rule-based evaluation results of different settings with respect to success rate@$k\%$ and coverage rate@$k\%$ on test set. Results show that increasing the resolution $r$ of grid corners generally improves the success rate of GPNet, depending on a proper size of 3D proposal space; given a fixed $r$, enlarging the 3D proposal space improves coverage at the sacrifice of reduced success rate. Overall, our proposed GPNet largely outperforms a naive version, and also outperforms the state-of-the-art 6-DOF GraspNet [20] under both measures, even when it uses a final, separate step of grasp refinement. The full curves of success rate vs coverage rate of GPNet ($r = 10, b = 22cm$) and 6-DOF GraspNet [20] are shown in Figure 3.

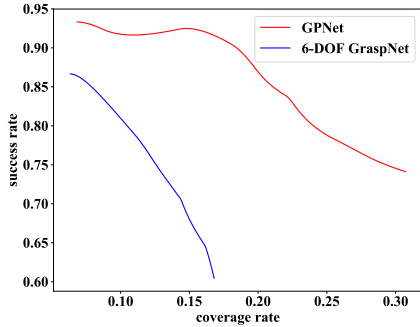

Figure 3: The curves of success rate vs coverage rate of GPNet and 6-DOF GraspNet.

To understand how results of GPNet depend on the size of training set and also the number of annotations per object, we conduct such experiments by either using reduced training sets, or reduced numbers of annotations per object. Table 2 reports simulation results in Pybullet based physics engine, which show that both of the investigated factors affect the performance of GPNet significantly. In fact, sufficiency of training samples with densely annotated grasps is crucial to achieve high success rates.

Table 1: Rule-based evaluation results (success rate@$k\%$ and coverage rate@$k\%$) on the test set. Results of GPNet variants with specific resolution $r$ in a 3D proposal space of length $b$ are reported.

| Methods | | success rate@$k\%$ | | | | coverage rate@$k\%$ | | | |
|---|---|---|---|---|---|---|---|---|---|
| | | 10 | 30 | 50 | 100 | 10 | 30 | 50 | 100 |
| 6-DOF GraspNet [20] | w/o refinement | 0.867 | 0.850 | 0.711 | 0.534 | 0.039 | 0.039 | 0.094 | 0.132 |
| | w/ refinement | 0.867 | 0.833 | 0.733 | 0.534 | 0.063 | 0.063 | 0.122 | 0.168 |
| GPNet-Naive | $r=10, b=22cm$ | 0.372 | 0.313 | 0.278 | 0.215 | 0.022 | 0.058 | 0.100 | 0.142 |
| GPNet | $r=3, b=22cm$ | 0.844 | 0.833 | 0.800 | 0.649 | 0.051 | 0.107 | 0.191 | 0.273 |
| | $r=7, b=22cm$ | 0.898 | 0.833 | 0.822 | 0.713 | 0.061 | 0.113 | 0.201 | 0.300 |
| | $r=10, b=22cm$ | **0.933** | **0.932** | 0.820 | **0.729** | 0.068 | 0.144 | 0.199 | 0.307 |
| | $r=10, b=10cm$ | 0.856 | 0.776 | 0.695 | 0.570 | 0.055 | 0.112 | 0.169 | 0.274 |
| | $r=10, b=30cm$ | 0.900 | 0.869 | **0.846** | 0.712 | **0.073** | **0.157** | **0.231** | **0.308** |

Table 2: Simulation results (success rate@$10\%$) on the test set when training GPNet with different numbers of grasp annotations per object and different ratios of the training set.

| # Avg. annotations per object | Accuracy | Ratio of training set | Accuracy |
|---|---|---|---|
| $10K$ | 0.650 | 1/4 | 0.522 |
| $50K$ | 0.730 | 1/2 | 0.700 |
| $100K$ | **0.900** | 1 | **0.900** |

## 5.2  Comparative Simulation Results

We conduct simulation test in Pybullet based physics engine. Results of GPNet in Table 3 suggest that to have successful grasps in practice, it is important to achieve diversity of grasp predictions by spreading the anchor grids in a properly sized 3D proposal space. Given the proper grid resolution and 3D proposal space, our method outperforms the state-of-the-art 6-DOF GraspNet [20]; note that the final, separate step of grasp refinement is essential for 6-DOF GraspNet to have high success rates; in contrast, our method gives good results in an end-to-end learning and prediction. In Table 3, our results are also better than those of planar grasp using the GQCNN model of DexNet [17]. Planar grasp is easier to learn but might be difficult to practically pick up objects of certain categories; this is reflected in the lowest success rate (in fact, zero success rate) of GQCNN on the category of *bowl*.

Table 3: Simulation-based evaluation results (success rate@$k\%$) on the test set. Results of GPNet variants with specific resolution $r$ in a 3D proposal space of length $b$ are reported.

| Methods | | $k=10$ | $k=30$ | $k=50$ | $k=100$ |
|---|---|---|---|---|---|
| GQCNN of planar grasp in DexNet [17] | | 0.783 | 0.742 | 0.663 | 0.464 |
| 6-DOF GraspNet [20] | w/o refinement | 0.433 | 0.367 | 0.311 | 0.207 |
| | w/ refinement | 0.800 | 0.594 | 0.508 | 0.354 |
| GPNet-Naive | $r=10, b=22cm$ | 0.100 | 0.095 | 0.083 | 0.054 |
| GPNet | $r=3, b=22cm$ | 0.644 | 0.637 | 0.561 | 0.371 |
| | $r=7, b=22cm$ | 0.767 | 0.711 | 0.656 | 0.557 |
| | $r=10, b=22cm$ | **0.900** | **0.761** | **0.723** | **0.588** |
| | $r=10, b=10cm$ | 0.494 | 0.433 | 0.393 | 0.253 |
| | $r=10, b=30cm$ | 0.833 | 0.702 | 0.679 | 0.574 |

## 5.3  Robot Experiment

It is a common anxiety that there exists a risk of domain discrepancy between simulated and real environments, e.g., due to camera and robot set-ups. To examine and verify performance of our predicted grasps in real world, we conduct grasp experiments with the models trained on our synthetic dataset and test in real scenario using a UR5 collaborative robot with Robotiq 2F85 gripper. Visualization of our grasp setups is given in the supplementary material. For grasps predicted by

Table 4: Comparative results of real robot test. The numbered objects are shown in the supplementary material.

| Object index | #1 | #2 | #3 | #4 | #5 | #6 | #7 | #8 | #9 | #10 | |
|---|---|---|---|---|---|---|---|---|---|---|---|
| GPNet | 2/3 | 3/3 | 3/3 | 3/3 | 3/3 | 3/3 | 3/3 | 2/3 | 3/3 | 2/3 | |
| 6-DOF GraspNet [20] | 2/3 | 2/3 | 3/3 | 2/3 | 1/3 | 0/3 | 2/3 | 3/3 | 1/3 | 3/3 | |
| Object index | #11 | #12 | #13 | #14 | #15 | #16 | #17 | #18 | #19 | #20 | Overall |
| GPNet | 3/3 | 2/3 | 2/3 | 3/3 | 3/3 | 2/3 | 3/3 | 0/3 | 3/3 | 3/3 | 85% |
| 6-DOF GraspNet [20] | 3/3 | 3/3 | 3/3 | 3/3 | 2/3 | 3/3 | 3/3 | 0/3 | 3/3 | 2/3 | 73% |

different methods, we discard those whose actuation paths are not obtained by motion planning. The only criterion to judge success of a grasp is that the robot elevates the object over 30cm and returns to its original state. We conduct 3 grasp trials per object, and report the averaged results. Table 4 shows that performance from both methods of 6-DOF grasping only drops slightly compared with simulated testing, and our GPNet outperforms the current state-of-the-art 6-DOF GraspNet [20]. We put more experiments into supplementary material with a video recording.

## Broader Impact

Traditional automation of robotic grasping requires availability of both object CAD models and annotations of grasp affordance. Recent data-driven solutions push the technologies towards a higher level of intelligence, by learning the grasp actuation directly from visual observations. However, most of these solutions are configured as 4 degrees of freedom, whose usage is restricted to a controlled planar grasp setting. The studied 6-DOF, visual grasp learning enables the technologies flexible enough to support a range of applications in industrial manufacturing, retail, and home assistance. However, the improved flexibility may cause downside impacts when the models are under adversarial attacks, causing failure of the automated system, and even life-threatening damage. Negative impact may also arise if the technology is abused. We encourage future research to mitigate these risks; we also expect policymakers would take active actions to penalize misuse of such technologies.

## Acknowledgements

This work is supported in part by National Natural Science Foundation of China (Grant No. 61771201), Program for Guangdong Introducing Innovative and Enterpreneurial Teams (Grant No. 2017ZT07X183), Guangdong R&D key project of China (Grant No. 2019B010155001).

## Footnotes

[3]Note that $(c_1, c_2)$ also determine an additional freedom of the width $w = \|c_2 - c_1\|$ of parallel-jaw gripper, which is only involved in training and inference of GPNet, but not used in evaluation of predicted grasps. Nevertheless, we still write $g = (c_1, c_2, \phi)$ or $g = (c_1, x, \phi)$ with a slight abuse of notation.

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
