[Supplementary Material]

# Supplementary material for "Grasp Proposal Networks: An End-to-End Solution for Visual Learning of Robotic Grasps"

## A    Illustration of Grasp Parameterizations

We present illustration of the 6 degrees-of-freedom (DOFs) grasp parameterization in Fig.1-(a). In Fig.1-(b), we illustrate terms that are used in computation of analytic score for a grasp candidate.

(a)                                                                    (b)

Figure 1: (a) Illustration of the 6-DoF grasp parameterization. The two contact points are denoted as $c_1$ and $c_2$, with the corresponding normal vectors $n_1$ and $n_2$. A grasp candidate is parameterized by gripper center $x$ and gripper orientation $\theta$. An additional freedom of gripper opening width $w$ is sometimes used in the literature. (b) Illustration for analytic score computation, where $z$ is the object mass center, $\rho_i$ is the vector from the torque origin (usually the mass center) to the $i$-th contact point.

## B    The Dataset Acquisition Framework

We present in this section our framework to produce synthetic object grasp dataset using physics engine of PyBullet [4]. Over the years the community has collected online repositories of categorized 3D object models [1, 6, 8]. Among them, the ShapeNetSem dataset [15] contains annotations of physical attributes such as material density and static friction coefficient, which are essential for use in our grasp annotation framework.

**The set-up of simulation environment** We consider a scenario of grasping in PyBullet a singulated object resting on a plane using parallel-jaw gripper. Given an object model in ShapeNetSem, we first normalize the model such that the longest side of its bounding box is smaller than $150\mathrm{mm}$ and the shortest side is larger than $60\mathrm{mm}$, while keeping the aspect ratio fixed. We then drop the model from a random position and orientation above the plane, and save the scene configuration (i.e., object

position and pose) once the object stably stands [1]. The saved scene configuration is independently used for two subsequent processes, i.e., simulation of grasp trials and rendering of depth (and RGB) images under random views. We use Blender [3] for image rendering.

**Sampling of grasp candidates** Our annotation generation starts with sampling parameterized grasp candidates for any object stably placed in a scene configuration, a process more involved than sampling simplified candidates as in [5]. Specifically, we first sample a contact point $\mathbf{c}_1 \in \mathbb{R}^3$ on mesh model of the object surface $\mathcal{S}$, and correspondingly sample a direction $\mathbf{v} \in \mathbb{S}^2$ uniformly at random from the friction cone [13] associated with $\mathbf{c}_1$ (more details in Appendix C); we then compute the contact point $\mathbf{c}_2 = \mathbf{c}_1 + \eta^* \mathbf{v}$ paired with $\mathbf{c}_1$, where $\eta^*$ is set to ensure that $\mathbf{c}_2$ is on $\mathcal{S}$; we choose to keep the pair $(\mathbf{c}_1, \mathbf{c}_2)$ by checking whether it satisfies the antipodal constraints of $\mathbf{v}^\top \mathbf{n}_1 \leq \cos(\arctan(\gamma))$ and $\mathbf{v}^\top \mathbf{n}_2 \leq \cos(\arctan(\gamma))$ [2], where $\mathbf{n}_1 \in \mathbb{S}^2$ and $\mathbf{n}_2 \in \mathbb{S}^2$ are surface normals respectively at $\mathbf{c}_1$ and $\mathbf{c}_2$, and $\gamma$ denotes static friction coefficient of the surface. Each kept pair of antipodal contacts determines the gripper center $\mathbf{x} = (\mathbf{c}_1 + \mathbf{c}_2)/2$, and the "roll" and "yaw" orientations of $\theta$ (and additionally the width $w$); we finally sample the left free parameter of "pitch" orientation to obtain a group of grasp candidates that share the same $(\mathbf{c}_1, \mathbf{c}_2)$. We use the above procedure to sample multiple pairs of contact points.

**Generation of grasp annotations** Given a grasp candidate, we call inbuilt PyBullet functions to plan and actuate the grasp. If the gripper does not collide with ground, we close the gripper until PyBullet detects any contacts where each finger of the gripper touches on the object; otherwise it is considered as a *collided grasp candidate*. If the contacts detected by PyBullet are almost identical to the pair of *antipodal contacts* from which the grasp candidate is generated, and the gripper opens to a width roughly the same as $w$, we step forward to lift the object to a certain height and move it around. The grasp candidate $\mathbf{g}$ is annotated as *success* if the object does not slide from the gripper and fall down during the lifting process; otherwise $\mathbf{g}$ is annotated as *failure*. In this work, we also compute an accompanying analytic score for each $\mathbf{g}$. Appendix C gives the details.

**Data clean-up** Grasp candidates in the preceding section are generally sampled uniformly at random from the space of contact pairs that satisfy antipodal constraints, which may produce a plenty of candidates non-reachable by the parallel-jaw gripper — the gripper may collide with either the ground or faces of object meshes. We remove those contact pairs too close to the ground, while leaving other non-reachable candidates to be dealt with by PyBullet's collision-handling functionality. Successful grasps usually concentrate on certain areas of an object surface. To remove nearly duplicate ones, for any two annotated grasps, we keep one of them when their gripper center positions are too close and grasp orientations are almost the same. This also makes obtained grasp annotations distribute more diverse over the object surface.

## C   Analytic Score Computation of Grasp Candidates

There exist different analytic metrics to evaluate robotic object grasps [11, 10, 14]. We use the Ferrari-Canny metric [7] in this work. Given a grasp candidate $\mathbf{g}$ that is specified by a contact pair $(\mathbf{c}_1, \mathbf{c}_2)$, we approximate the fiction cone at $\mathbf{c}_i$, $i \in \{1, 2\}$, into $L$ facets, as illustrated in Fig.2, with the set of vertices defined as

$$\mathcal{F}_i = \{\mathbf{n}_i + \gamma \cos(\frac{2\pi j}{L})\mathbf{t}_{i,1} + \gamma \sin(\frac{2\pi j}{L})\mathbf{t}_{i,2} \mid j = 1, \ldots, L\}, \tag{1}$$

where $\mathbf{t}_{i,1} \in \mathbb{S}^2$ and $\mathbf{t}_{i,2} \in \mathbb{S}^2$ are the two tangent vectors at $\mathbf{c}_i$. Given object mass center $\mathbf{z}$, each force vector $\mathbf{f}_{i,j} \in \mathcal{F}_i$ exerts a corresponding torque $\tau_{i,j} = c(\rho_i \times \mathbf{f}_{i,j})$, where $\rho_i = \mathbf{c}_i - \mathbf{z}$ and $c$ is an adaptive parameter to make the torque invariant to scale of the object. Appending the corresponding force and torque gives the wrench $\mathbf{w}_{i,j} = [\mathbf{f}_{i,j}^\top, \tau_{i,j}^\top]^\top \in \mathbb{R}^6$. Under the soft finger model [17], we add an extra wrench vector $\mathbf{w}_{i,L+1} = [\mathbf{0}^\top, \mathbf{n}_i^\top]^\top \in \mathbb{R}^6$, and construct the grasp wrench space (GWS) [7] as

$$\mathcal{W} = \text{ConvexHull}\left(\cup_{i=1}^2 \{\mathbf{w}_{i,1}, \ldots, \mathbf{w}_{i,L+1}\}\right). \tag{2}$$

Let $s(\mathbf{g})$ be the analytic score to be annotated. The grasp $\mathbf{g}$ is not *force closure* when $\mathcal{W}$ does not contain the origin of 6-dimensional wrench space, i.e., $\mathbf{0} \notin \mathcal{W}$, and we annotate $s(\mathbf{g}) = -1$.

Conversely, $\mathbf{g}$ is considered as force closure and we compute the analytic score by the distance between the origin and the nearest facet of $\mathcal{W}$ [7], namely radius of the maximum hypersphere centered at the origin and contained in $\mathcal{W}$

$$s(\mathbf{g}) = \arg\max_\varepsilon \mathcal{B}(\varepsilon)$$
$$\text{s.t. } \mathcal{B}(\varepsilon) = \{\mathbf{w} \in \mathbb{R}^6 \mid ||\mathbf{w}||^2 < \varepsilon\}, \ \mathcal{B}(\varepsilon) \subseteq \mathcal{W}. \tag{3}$$

(a)                    (b)

Figure 2: (a) The force $\mathbf{f}$ exerted at contact $\mathbf{c}$ must lie within the cone otherwise it will cause slippage. (b) In order to simplify the grasp analysis process, we approximate the friction cone with a sided pyramid. $\mathbf{f}$ is a convex combination of $\{\mathbf{f}_0, ..., \mathbf{f}_8\}$

# D   Additional Material for Robot Experiments

Figures 3a and 3b show the 20 objects to be grasped and our grasp setup. We also compare the prediction time of the existing methods as shown in Table 1. Note that the results are evaluated based on generating $10k$ grasps per object.

Table 1: Computational analysis.

| Methods | GPD[16] | 6-DOF GraspNet [12] | GPNet |
|---------|---------|---------------------|-------|
| Time(s) | $> 10$ | $\sim 10$ | $\sim 2.0$ |

(a) Objects for real test                    (b) Robot disposition

Figure 3: Real test setting.

## Footnotes

[1]Since PyBullet cannot directly load concave triangle mesh models as collision into simulation environment, we use Hierarchical Approximate Convex Decomposition [9] to decompose each of concave mesh models in ShapeNetSem as several convex components.