[Reviews · NeurIPS 2020]

Review 1

Summary and Contributions: The papers proposes learning a grasp proposal network for 6-DOF grasping, generated on a synthetic dataset of objects in simulation. The task focused on is grasping a singulated object on a flat surface with a parallel-jaw gripper. The main distinction from prior grasp annotation / grasp proposal methods is that they aim for 6-DOF grasping rather than planar grasps. For each object, a large number of grasps are attempted in simulation. This creates a large set of grasp annotations, and a neural net is trained to take the object point cloud and generate candidate grasps with high success rate. The network works by partitioning 3D space into r x r x r grid cells, treating each grid cell as an anchor for the grasp center. For each anchor, the network predicts an angle and offset to apply. At training time, the grasp proposals are filtered to only be ones near the ground-truth grasps and objects, to reduce training needs. The proposals are then scored based on the annotation label. The models are then evaluated according to 3 methods 1. Error to the ground-truth grasp annotation 2. Grasp performance in sim 3. Grasp performance in real

Strengths: The generated dataset sounds useful for pushing 6-DOF grasping. The claims are all very reasonable, and their results suggest that the anchor-based architecture makes the problem easier to learn. The work is very thorough in describing its dataset generation and network architecture.

Weaknesses: The paper primarily feels like a dataset paper. There is nothing inherently wrong with this, but there have been several grasp datasets in prior work, including 6-DOF grasp datasets from 6-DOF GraspNet. The difference between that dataset and this one is that this dataset is 22.6M grasps instead of 10.8M grasps, and the grasps are over a slightly wider set of ShapeNet objects (226 objects instead of 206 objects, 8 categories instead of 6 categories). So, on the dataset side, the question I'm considering is how useful this 2x larger generation is. Given these are simulated grasps, which are less costly to generate, my intuition is that this dataset is useful but not significantly so compared to existing ones. If there is an argument for why the 2x larger dataset is a big deal, I could change my mind here. As for the anchor-based architecture, I think the idea of using models based on 2D object detection is a good idea, and the results for it seem good, but I'm very suspicious of the GPNet-Naive results. This row of the table suggests that removing the anchor coordinates from the grasp proposals completely ruins performance. This seems like too strong a result, given that prior baselines still have reasonable performance on this dataset. Could the authors explain why the naive performance is so bad? I also do not understand why r,b are still mentioned for GPNet-Naive - if the anchor centers are not given as features, then grasps proposed in each of the r x r x r grid cells are not distinguishable, and if those aren't distinguishable, then the baseline feels worthless.

Correctness: The claims look correct to me.

Clarity: Some wording issues, like "We totally obtain" should be "in total we obtain", but otherwise the paper was clear.

Relation to Prior Work: The paper mentions relevant prior work, but is not very clear on the difference between prior work and this work, especially when comparing to prior 6-DOF synthetic datasets (I believe the difference between this work and planar grasp work is fairly clear.) The visual grasp learning section would be better if it explained how their anchor-based architecture differs from prior work.

Reproducibility: Yes

Additional Feedback: This paper feels like much more of a robotics paper than a NeurIPS paper. I feel that overall this paper is valuable and written well but I'm not sure it adds much to prior work. Edit: I have read the other reviews and the rebuttal. Thank you for the explanation about GPNet-Naive. I feel the authors have addressed some of my concerns about the dataset, but I'm not sure if antipodal labels are an important enough distinction, so I plan to keep the same score.


Review 2

Summary and Contributions: Generating a set of grasp proposals for various objects is a tough problem and is an interest in the field of robotics. The paper proposes a network to deal this problem, Apart from generating a set of proposals, it also gives confidence for each proposal. Specifically, the authors uses Pointnet++ to encode object shape features on each surface point, then feed all pairs of grip centers and one surface point as initial proposals to three headers to do antipodal-based proposal pruning, complete grasp parameter generation and confidence evaluation. The training data is generated using physics simulation. And the proposed method has been evaluated on different experiments, showing state of the art performance.

Strengths: The proposed method is able to generate large set of grasp proposals along with confidence. Experiments show the state of the art performance of the proposed method. This work should be the interest of some people in the robotic field.

Weaknesses: 1. Lack of theoretical novelty. It's a task-specific solution and has a limited impact on other areas. This could potentially be a high impact paper on robotic journal or conference though. 2. Though an antipodal-based classifier can prune most of the proposals, pruning and evaluating every possible combination of gripper center and surface point is not very efficient. It could be better if only promising proposals are brought up at the beginning and go through further processing. For example, an environment-and-object based surface point proposal will be more interesting.

Correctness: Overall the method and claims are valid. Please refer to the first point in the Additional Feedback part.

Clarity: Yes

Relation to Prior Work: Yes. And if it is inspiring, I suggest the authors to involve some human hand grasping works such as 'ContactDB: Analyzing and Predicting Grasp Contact via Thermal Imaging'.

Reproducibility: Yes

Additional Feedback: 1. l.109 the rule used for the gripper angle is not very convincing. To measure the distance between two unit quaternions p and q, I suggest something like either pq* or the length of shortest path on 4D unit sphere between p and q. 2. Minor. In Fig 1, the FIRST header Antipodal Classifier is on the bottom right corner, which is not intuitive and may cause certain confusion if not reading in details. It could be better if the three headers can be arrange from top to bottom or from left to right rather than from bottom to top.


Review 3

Summary and Contributions: The paper tackles the task of 6DOF grasping. It proposes a grasp proposal network (GPNet) that uses PointNet++ style architecture to classify grasp proposals (at grasp anchor locations), and regress to the 6DOF grasp parameters. The paper conducts experiments on datasets derived out of the ShapeNet dataset, and on a real robotic platform.

Strengths: 1. Paper tackles an important problem, that of 6DOF grasping. 2. Paper designs a new architecture for 6DOF grasp prediction. It adopts the standard object detection paradigm, and proposes to score grasp candidates (and additionally output grasp angles, and offsets). Grasp candidates are predicted from a set of densely sampled anchor locations. The grasp proposal network uses the PointNet++ architecture. Overall, I think the proposed architecture is novel, and I haven't seen specifically this architecture being used for 6DOF grasp prediction. The specific formulation makes intuitive sense. 3. The paper constructs a dataset using objects from the ShapeNet dataset for training, and evaluating the proposed Grasp Proposal Net. Paper reports favorable comparison to recent work on this problem [23].

Weaknesses: 1. While the exact architecture may not have been tried on this problem (or even otherwise), the architecture builds upon existing ideas in the field (casting grasp prediction as grasp proposal classification [24]), sim2real for grasping using depth data [17], uses of anchors [30], PointNet++[27]). Thus, I won't consider the novelty of the proposed architecture super high. 2. At the same time, experimental evaluation could be stronger. Given limited novelty, I would expect a through empirical evaluation of the different aspects of the proposed model, eg: is PointNet++ style architecture necessary, or would a more natural choice 3D CNN (or 2D Depth CNN as in DexNet) work better. 3. Why GPNet-Naive only report r = 10, b = 22, while a more through validation has been done for GPNet? 4. Use of non-standard evaluation metrics. Past work in [23] seems to use plots of success vs coverage. Current paper only reports 4 points on this curve. Why not include plots similar to ones in [23]? 5.Past papers in learning based grasping (albiet in 2D) employ a more thorough real experimental evaluation. By that measure the real world evaluation is weak.

Correctness: Empirical methodology is correct, though could me more thorough in a) equivalent validation for baselines (see 3. in Weaknesses), b) use of standard metrics as used in past work, c) comparison to more ablations and baselines.

Clarity: The paper doesn't clearly identify its contribution. L48 to L58 mention that the proposed GPNet is novel, but fails to crisply identify what individual aspects are novel and in what way.

Relation to Prior Work: No, the paper does not clearly establish relationship to past work. It does not identify what aspects of the proposed approach are novel.

Reproducibility: No

Additional Feedback: == After Rebuttal == Thanks for your response, and providing additional experiments.


Review 4

Summary and Contributions: The focus of the work is computing 6DOF grasps for a parallel jaw gripper over point clouds of previously unseen objects. With respect to related literature, the claim is that this approach can compute a relatively more diverse set of grasps on objects. This is based on a grasp proposal module that defines anchors of grasp centers at regular 3d grid positions.

Strengths: - The motivation of generating a diverse set of robust grasp is very relevant as diverse grasps are useful for high-level task planning and to increase the chance of success due to kinematic constraints and collisions. The evaluation is appropriately designed to back the claim. - The idea of using 3d grid centers as anchors for regressing grasps over point cloud is novel (although derived from similar ideas in 2d scenarios and in object detection). - The design of grasp pruning -> grasp regression -> grasp classification with feature extraction neighborhood based on grasp proposal is novel.

Weaknesses: See Additional feedback and questions section.

Correctness: Yes

Clarity: Yes

Relation to Prior Work: Yes

Reproducibility: Yes

Additional Feedback: - The advantage of this approach over [33][34] as mentioned in Line 40 is mostly computational. However, no computational analysis is done to support this claim. Do these approaches achieve a diverse set of robust grasps when given enough time - how much time does it take. The code for these approaches is publicly available. - Why do the grasps from [23] tend to focus on a single-mode? Is there a theoretical limitation of the approach? and how does the proposed approach address it? It might be more insightful to elaborate on this. - The grasp proposals are first pruned based on antipodal criteria and then passed on to the regression module. Given the grasps are already antipodal (within a threshold) how important is the regression module? (especially for high-resolution scenarios) 1. Is the main objective of this module to estimate the free rotation parameter? 2. Is it only useful for low-resolution scenarios? In that case, what is the computational benefit of this over high resolution? More discussion regarding this aspect in the experiment section could be insightful. - The training and test data are from the same object categories. So it is not a fair claim to say that the learning can be generalized to completely unseen objects. This should be specified in the introduction. - It was claimed in the paper that the method is flexible to support either a more precise or more diverse grasp by focusing or spreading the anchors of grasp centers in 3D space. This claim was not backed empirically. ****************** After Rebuttal ****************** Appreciate the response and I think these discussions would be very useful in the revised version of the paper.

[Author Response · NeurIPS 2020]

We thank all reviewers for their constructive comments. Due to space limit, we address the major concerns as follows.

**Synthetic dataset (R1)** - We note that many of the (synthetic or real) datasets prepared in robotic grasp learning research are not made fully public. For example, configuration of the simulation environment used in the closely related work of 6-DOF GraspNet [23] is not publicly available, which makes it difficult to directly and fairly compare different methods. This is the main reason why we have to prepare our own synthetic data. In terms of additional features of our dataset other than the doubled size, more categories and instances, a valuable one is that our dataset has antipodal label for each grasp sampling, which supports physically sensible models such as GPNet. To contribute to the community, we will make all our dataset, including configuration of simulation environments, publicly available.

**Results of GPNet-Naive (R1)** - In GPNet-Naive, we still use anchor coordinates in the grasp proposal phase, but not for grasp regression. The grasp proposals are distinguishable because we use an anchor-dependent method for feature extraction (Sec. 4.2). Without anchor coordinates as input features, the Antipodal Classifier does not work well and the regression loss is much higher than that of GPNet. We will improve the description of GPNet-Naive in the paper.

**Novelties of GPNet (R1, R2, R3)** - To our best knowledge and as R4 said, our work is the first one that uses 3D anchors to generate 6-DOF grasps. Our grasp parametrization using surface contact, grasp center, and 'pitch' angle (L149) enables definition of anchor based grasp proposals, which is intuitive and physically sensible. With this parametrization and grasp proposal module, we only need to regress 4 parameters of center offset and 'pitch' angle, instead of 6 ones as in [23]; reduction of variables makes learning much easier. Our novelty also presents in design of antipodal validity -> grasp regression -> grasp classification (cf. Fig.1), where we prune grasp proposals by antipodal constraint and take the regressed centers and angles as input when scoring the grasp candidates, while [23] does not consider antipodal constraint, and [4,22,28] output grasp candidates and their confidences in a parallel way, which is suboptimal for ranking grasps (L211-L213). In addition, our anchor-dependent feature extraction differs from prior works, and our GPNet is trained end-to-end, while [23] trains its Grasp Sampler and Grasp Evaluator independently.

**Pruning in the beginning (R2)** - Thanks. We have in fact tried an affordance detection module in GPNet, where contact points suggested by affordance detection are picked for grasp proposal and the rest ones are removed. On average, $32\%$ prediction time is saved with no sacrifice of success rate. We will include these results in the paper.

**Rule-based evaluation (R2)** - We measure the rotation difference by $\Delta_\theta = 2 \arccos \mathbf{q}_1 \cdot \mathbf{q}_2$. For ease of understanding, we formulate it as L109. We have tested both and there is no much difference. We will revise the text accordingly.

**More empirical evaluation on model variants (R3)** - Thanks for the suggestion. We have in fact ever tried other modeling choices such as 2D Depth CNN: we use 2D Depth CNN and 2D grids (2D version of GPNet) to generate 6-DOF grasps, but it only produces $13.4\%$ success rate@$10\%$ and $4.3\%$ coverage rate@$100\%$, much worse than results of GPNet in Tables 1 and 3. For other choices such as voxelizing point cloud as input of 3D CNN, we note that it would sacrifice precision of 3D location, as verified by recent methods on KITTI benchmark of 3D object detection; it is harmful for precise regression of grasp pose as well. We will include these additional evaluations in the paper.

**More experiments on GPNet-Naive (R3)** - We just pick one GPNet-Naive variant to show the importance of anchor coordinates as input for antipodal classification and grasp regression. Success rates@$10\%$ of GPNet-Naive are respectively $0.050, 0.119, 0.100, 0.061, 0.053$ when $(r, b) = (3, 22cm), (7, 22cm), (10, 22cm), (10, 10cm), (10, 30cm)$ .

**Standard evaluation metrics (R3)** - Without losing fairness, we have selected 4 representative points on the curve of success rate vs coverage rate to report in the paper. A full curve is shown as the figure right; AUCs (Area Under Curves) of GPNet and 6-DOF GraspNet are $0.206$ and $0.081$, respectively. Note that we use a stricter criterion for coverage rate, where both center and rotation distances are considered (only center distance is considered in [23]).

**Real-world evaluation (R3)** - Thanks for the suggestion. Although our setting of real-world experiments is almost identical to that in [23], we will pursue more thorough real experiments in future research.

**Computational analysis (R4)** - Computations of existing methods are evaluated based on generating $10k$ grasps per object. Our GPNet takes $\sim 2.1s$ from input to scoring the generated grasps, while GPD [33] takes $> 10s$ for grasp sampling and grasp classification, and 6-DOF GraspNet takes $\sim 10s$ from grasp generation to grasp refinement.

**Single-mode issue of 6-DOF GraspNet [23] (R4)** - Grasp proposal of [23] is based on a trained variational auto-encoder (VAE). VAE suffers from an issue of "latent variable collapse" [Dieng et al.,AISTATS'19]. The posterior collapses to a simple prior and inference network fails to learn good representations, especially if the likelihood model is powerful, e.g. as in [23]. Thus, directly sampling the latent vector over normal distribution may make predicted grasps concentrate on a single mode. The lower coverage rate of 6-DOF GraspNet in Table 1 also supports this analysis.

**Importance of regression module (R4)** - Though we prune the grasp proposals based on antipodal criterion, there is no guarantee that the remaining ones are precise enough to be successful, especially for low-resolution scenario. The regression module is to regress a more precise grasp center and free rotation. If increasing the grid resolution for an improved precision, both the number of grasp proposals and the time of antipodal pruning will increase cubically.

**Empirical evaluation of focusing or spreading anchors (R4)** - In Table 1, we have shown that best coverage rate (more diverse) is obtained when spreading the anchors ($r = 10, b = 30cm$), and best success rate (more precise) is obtained when focusing the anchors ($r = 10, b = 22cm$). We will improve the description in the paper.

**Intra-category generalization to unseen objects (R4)** - Thanks and we will specify this in the introduction.

[Meta-Review · NeurIPS 2020]

This paper proposes an approach to predict multiple stable 6-dof grasp parameters for standard parallel-jaw grippers from object point cloud inputs, with associated confidence values. Grasps are represented as tuples of (contact points of the 2 jaws and the pitch angle of the gripper), which motivates the new architectural choices proposed here, inspired by standard architectures in 2D object detection. While the network is trained end-to-end, it is internally decomposed in a sensible stage-wise manner. They also create a synthetic 22.6M 6-DOF grasp dataset built on ShapeNet objects using physics simulation, which upon public release, will be the largest such dataset. Finally, there are some limited transfer results that demonstrate transferability to real-world grasping with acceptable performance drop. The author responses were helpful in addressing many of the weaknesses raised in the first round of reviews (and I urge the authors to incorporate key response points into future versions), but a few concerns remain. In particular, grasping in robotics is a long-studied field, typically involving extensive real-world experimentation and benchmarking for proper evaluation. By comparison, the real-world transfer results presented here are limited. While I am recommending acceptance, I urge the authors to consider more extensive experimentation on reporting real-world grasping results. Another drawback is that, unlike a lot of recent grasping work, the method is only ever evaluated on objects from the training categories. It would be interesting to see evaluation on unseen object categories too. Finally, the current broader impact statement could be improved with some more careful thought, particularly about ethical consequences. "Negative impact may arise if the technology is abused" really doesn't say anything at all.